# Genome-wide association study of Buruli ulcer in rural Benin highlights role of two LncRNAs and the autophagy pathway

Jeremy Manry [1,2,9 ✉], Quentin B. Vincent[1,2,9], Christian Johnson[3,4], Maya Chrabieh[1,2], Lazaro Lorenzo[1,2], Ioannis Theodorou[5], Marie-Françoise Ardant[3,6], Estelle Marion [7], Annick Chauty[3,6], Laurent Marsollier[7], Laurent Abel[1,2,8] & Alexandre Alcaïs [1,2 ✉]

Buruli ulcer, caused by *Mycobacterium ulcerans* and characterized by devastating necrotizing skin lesions, is the third mycobacterial disease worldwide. The role of host genetics in susceptibility to Buruli ulcer has long been suggested. We conduct the first genome-wide association study of Buruli ulcer on a sample of 1524 well characterized patients and controls from rural Benin. Two-stage analyses identify two variants located within LncRNA genes: rs9814705 in *ENSG00000240095.1* ($P = 2.85 \times 10^{-7}$; odds ratio = 1.80 [1.43–2.27]), and rs76647377 in *LINC01622* ($P = 9.85 \times 10^{-8}$; hazard ratio = 0.41 [0.28–0.60]). Furthermore, we replicate the protective effect of allele G of a missense variant located in *ATG16L1*, previously shown to decrease bacterial autophagy (rs2241880, $P = 0.003$; odds ratio = 0.31 [0.14–0.68]). Our results suggest LncRNAs and the autophagy pathway as critical factors in the development of Buruli ulcer.

[1] Laboratory of Human Genetics of Infectious Diseases, Necker Branch, Institut National de la Santé et de la Recherche Médicale (INSERM) UMR 1163, Paris, France. [2] Université de Paris, Imagine Institute, Paris, France. [3] Fondation Raoul Follereau, Paris, France. [4] Centre Interfacultaire de Formation et de Recherche en Environnement pour le Développement Durable. Université d'Abomey, Calavi, Benin. [5] Center for Immunology and Infectious Diseases, INSERM UMR S 1135, Pierre and Marie Curie University, and AP-HP Laboratoire d'Immunologie et Histocompatibilité Hôpital Saint-Louis, Paris, France. [6] Centre de Dépistage et de Traitement de la Lèpre et de l'Ulcère de Buruli (CDTLUB), Pobè, Benin. [7] INSERM UMR-U892 and CNRS U6299, team 7, Angers University, Angers University Hospital, Angers, France. [8] St Giles Laboratory of Human Genetics of Infectious Diseases, Rockefeller Branch, Rockefeller University, New York, NY, USA. [9] These authors contributed equally: Jeremy Manry, Quentin B. Vincent. ✉email: jeremy.manry@inserm.fr; alexandre.alcais@inserm.fr

uruli ulcer is a chronic infectious disease caused by *Myco-bacterium ulcerans*. It received little attention until about 15 years ago, despite being more prevalent than tuberculosis or leprosy in some areas[1]. Interest in this disease was stimulated by the identification of Buruli ulcer as a neglected emerging tropical disease and the launch of the Global Buruli Ulcer Initiative by the World Health Organization (WHO) in 1998[2]. Buruli ulcer is now recognized as the third most frequent human mycobacterial disease worldwide, after tuberculosis and leprosy[3]. It occurs mostly in rural areas of tropical countries, and West Africa is considered the principal endemic zone[4]. Prevalence estimates between 2007 and 2016 ranged from 0.32 per 1000 [95% confidence interval (CI): 0.31–0.33] in Ivory Coast to 2.99 per 1000 [95%CI: 2.35–3.07] in Benin[5]. In 2018, the WHO reported a global increase in new cases of 39% relative to 2016[4].

Buruli ulcer is a devastating necrotizing skin infection characterized by pre-ulcerative lesions (nodules, plaques, and edema) that may develop into deep ulcers with undermined edges potentially extending to bones and joints. *M. ulcerans* induces painless skin necrosis through the production of mycolactone, a toxin that has been shown to play a key role in bacterial virulence and analgesia[6–8]. The painless nature of the disease often delays diagnosis, leading to late treatment and permanent disabilities, which affect more than 20% of patients, mostly children[9]. Considerable variability has been reported in the clinical presentation of Buruli ulcer, including spontaneous healing in both humans[10,11] and specific mouse strains[11]. This observation, together with the familial clustering of cases[12,13], support the view of a substantial contribution of host genetic factors to the response to infection with *M. ulcerans*.

This hypothesis is consistent with the discovery of rare inborn errors of immunity conferring a predisposition to severe infections with weakly virulent mycobacteria, such as the BCG vaccine, in the context of the syndrome of Mendelian susceptibility to mycobacterial diseases (MSMD)[14,15]. Rare genetic defects have also been shown to underlie severe forms of tuberculosis[16,17], and homozygosity for a missense polymorphism of *TYK2* was recently identified as the most common monogenic cause of tuberculosis[18,19]. A similar monogenic contribution to Buruli ulcer is supported by the recent identification of a microdeletion on chromosome 8 in a familial form of severe Buruli ulcer[20]. Genome-wide association studies (GWAS) have identified several common variants associated with leprosy[21–23], while this kind of variants seems to play only a limited role in tuberculosis[24]. By contrast, the role of common host genetic variants in the development of Buruli ulcer has been investigated in only three candidate-gene studies. These studies explored a total of seven genes selected on the basis of their involvement in MSMD, leprosy or tuberculosis. Significant association with Buruli ulcer was reported for common variants of six of these genes: *SLC11A1*[25], *PRKN*, *NOD2*, and *ATG16L1*[26] and *iNOS* and *IFNG*[27]. We performed a two-stage case-control GWAS, to investigate comprehensively the role of common variants in the development of Buruli ulcer, on a sample of 1524 individuals recruited at the Centre de Dépistage et de Traitement de la Lèpre et de l'Ulcère de Buruli (CDTLUB) of Pobè, Benin. Our analysis identifies two novel variants independently associated with the onset of Buruli ulcer and confirms the protective effect of a missense variant in the bacterial autophagy gene *ATG16L1*.

## Results

**Genome-wide analyses**. GWAS was performed on the discovery sample of 402 Buruli ulcer cases (sex ratio: 0.79; median age: 11 years) and 401 exposed controls (sex ratio: 0.72; median age: 40 years) living in villages in the Ouémé and Plateau *départements* (Benin) in which Buruli ulcer is endemic (Table 1; Fig. 1). Principal component analysis (PCA) on genotyped variants with a minor allele frequency (MAF) > 0.05 revealed no evidence of population stratification in our sample, either at the global (Supplementary Fig. 1a) or African-specific (Supplementary Fig. 1b) level as all our individuals clustered with those of the Yoruba population of Ibadan, Nigeria (YRI) and the Esan population of Nigeria (ESN) from the 1000 Genomes Project. Refined PCA performed only on individuals from our study ruled out cryptic stratification as cases and controls were evenly distributed across the two first components (Supplementary Fig. 1c). Following imputation, 10,014,109 high-quality autosomal variants were tested for association with two different phenotypic definitions of Buruli ulcer under the assumption of an additive genetic model.

We first analyzed the binary 'case/control' Buruli ulcer phenotype. Consistent with the refined PCA analysis, the quantile–quantile (Q–Q) plot (Supplementary Fig. 2a) showed no substantial deviation between the observed and the expected, i.e. asymptotic, distribution of the test statistics at the genome-wide level (genomic inflation factor $\lambda = 1.027$). The results of the GWAS tests are summarized in a Manhattan plot (Fig. 2a). In total, 517 variants had $P$ values $< 5 \times 10^{-5}$, including 10 with $P$ values $< 10^{-6}$. The most significant signal for association was observed for an imputed SNP (rs7246288) located on chromosome 19 with a $P$ value $= 2.80 \times 10^{-7}$, a MAF of 0.09 and an odds ratio (OR) estimated at 2.33 [95%CI: 1.62–3.34] for developing Buruli ulcer for TT vs. CT or CT vs. CC carriers. After linkage disequilibrium (LD) pruning (see the "Methods" section for details), 66 promising variants (i.e. displaying $P$ value $< 5 \times 10^{-5}$ and $10^{-6}$ for genotyped and imputed variants, respectively) were retained for genotyping and association testing in the replication sample (Supplementary Data 1).

Age at diagnosis has been shown to provide additional information about the architecture of the genetic contribution to complex diseases[28]. We, therefore, performed a GWAS taking

**Table 1 Distribution of sex and age according to the case/control status in the discovery and replication samples.**

| Covariate | | Discovery sample | | Replication sample | | Combined | |
|---|---|---|---|---|---|---|---|
| | | Cases | Controls | Cases | Controls | Cases | Controls |
| Sex | Male | 177 (44.0)[a] | 168 (41.9) | 198 (43.5) | 98 (41.1) | 375 (43.8) | 266 (41.6) |
| | Female | 225 (56.0) | 233 (58.1) | 257 (56.5) | 140 (58.9) | 482 (56.2) | 373 (58.4) |
| | Total | 402 | 401 | 455 | 238 | 857 | 639 |
| Age[b] | Mean age (SD) | 18.5 (16.7) | 39.8 (17.3) | 22.4 (17.3) | 26.1 (18.7) | 20.5 (17.1) | 34.6 (19.0) |
| | Median age | 11 | 40 | 15 | 22.5 | 13 | 35 |

SD standard deviation.
[a]Values are expressed as number (%).
[b]Age is given in years and corresponds to the age at the time of recruitment for controls and to the age at diagnosis for cases.

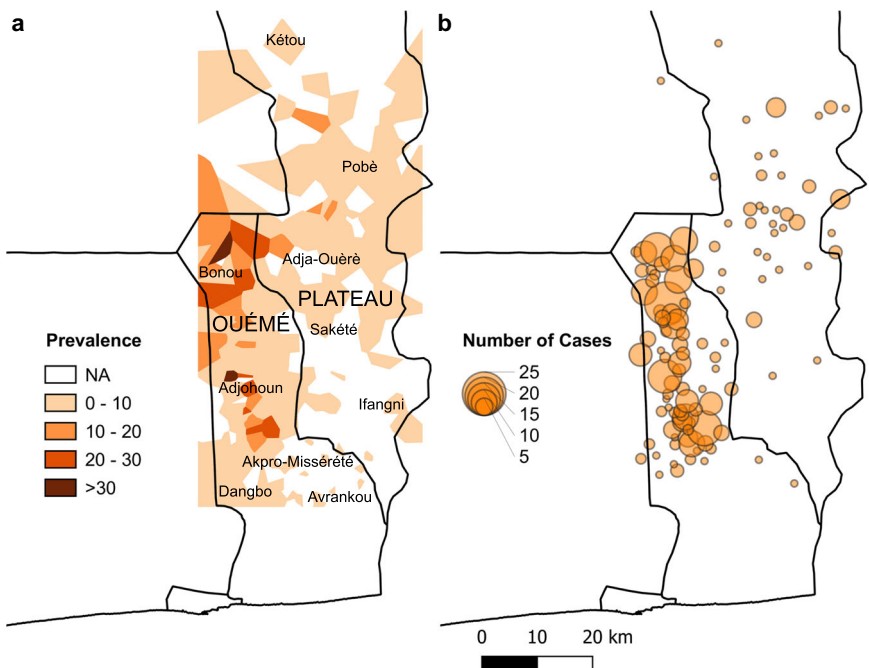

**Fig. 1 Geographic distribution of Buruli ulcer in Plateau and Ouémé in Benin. a** Prevalence calculated from cases registered between 2005 and 2012 in the *départements* of OUÉMÉ and PLATEAU (upper case) and subsequently refined at municipality level (lower case); **b** Number of cases per village included in the discovery sample used for the GWAS.

age at diagnosis of Buruli ulcer into account by means of survival analytic tools (see the "Methods" section). As expected, Q–Q plot showed no substantial deviation ($\lambda = 1.023$; Supplementary Fig. 2b). The results of the GWAS tests are summarized in a Manhattan plot (Fig. 2b). In total, 534 variants had a $P$ value < $5 \times 10^{-5}$, including nine with a $P$ value < $10^{-6}$. The most significant signal for association was observed for a genotyped SNP (rs34060873) located on chromosome 3 with a $P$ value = $1.99 \times 10^{-7}$, a MAF = 0.10 and a hazard ratio (HR) estimated at 0.51 [95%CI: 0.39–0.67] for the development of Buruli ulcer for AA *vs.* CA or CA *vs.* CC carriers. Applying the same strategy as the one used for the binary phenotype, 68 variants were retained for genotyping and association testing in the replication sample (Supplementary Data 1). Among the 66 and 68 variants retained from the analysis of Buruli ulcer per se and the one taking age at onset into account, respectively, 29 displayed $P$ values below our thresholds of selection in both analyses. For the replication analysis, the phenotypic definition providing the best $P$ value for these variants was used. Altogether, analysis of the two phenotypic models led to the selection of 105 independent variants, 99 of which being genotyped and 6 being imputed (Supplementary Data 1).

**Replication study**. In the second phase of our study, these 105 variants were genotyped in an independent replication sample composed of 455 cases (sex ratio: 0.77; median age: 15 years) and 238 controls (sex ratio: 0.70; median age: 22.5 years) (Table 1). After quality control, five variants were excluded (see the "Methods" section) and 100 variants were retained for association testing in the replication sample with the same phenotypic model (binary or age at onset) and the same allelic effect as observed for the discovery sample (Supplementary Data 1). Out of the 100 high-quality variants tested for association with Buruli ulcer in this replication sample, rs9814705 and rs76647377 displayed significant evidence of replication, i.e. one-tailed $P$ value of the association test < 0.01 (Supplementary Data 1 and Table 2).

The first evidence of replication was observed for the binary phenotype and rs9814705 on chromosome 3, for which the replication $P$ value was $6.51 \times 10^{-4}$. As this variant had been imputed in the discovery sample (imputation info value = 0.933), we decided to subject it to Sanger sequencing in the discovery sample. This resequencing effort slightly increased the discovery $P$ value from $3.71 \times 10^{-5}$ (Supplementary Data 1) to $1.67 \times 10^{-4}$ (Table 2). When merging our two samples, the combined $P$ value for the association between rs9814705 and Buruli ulcer was $2.85 \times 10^{-7}$. The minor allele C (MAF = 0.14) was the risk allele, and the OR for developing Buruli ulcer for CC vs. TC or TC vs. TT carriers was estimated at 1.80 [95%CI: 1.43–2.27] (Table 2). The proportion of Buruli ulcer patients was 53% for TT carriers, and 76% for CC carriers (Fig. 3, Supplementary Table 1). We then searched for additional variants in strong LD, i.e. $r^2 > 0.5$, with rs9814705, using the 1000 Genomes Phase III data for the YRI population. Three SNPs were identified: rs1513419 ($r^2 = 0.87$, GWAS $P$ value = $4.37 \times 10^{-5}$), rs7637582 and rs7615284 ($r^2 = 0.69$, and GWAS $P$ value = $1.01 \times 10^{-5}$ for both) (Fig. 4). To confirm the hypothesis of a single signal driving the association with the onset of Buruli ulcer in this chromosomal region, we performed association tests with all local variants conditioning on rs9814705 (i.e., rs9814705 was considered as a covariate in the analysis of the other variants). No variant in LD ($r^2 > 0.2$, shown in Fig. 4) with rs9814705, including the three abovementioned SNPs, displayed $P$ value < 0.05 consistent with a single association signal accounted for by rs9814705 in this region (Supplementary Fig. 3a). It is important to note that this cluster of four SNPs in high LD ($r^2 > 0.5$) spans 13.4 kb in a single intron of ENSG00000240095.1, a LncRNA of presently unknown function located on chromosome 3 with the closest identified protein-coding gene (*PLOD2*) located 107 kb away. While we cannot decide for the causal variant on the basis of $P$ values, our results pinpoint this LncRNA as a solid candidate for further investigations.

The second evidence of replication was observed for the age of onset phenotype and rs76647377 on chromosome 6, for which

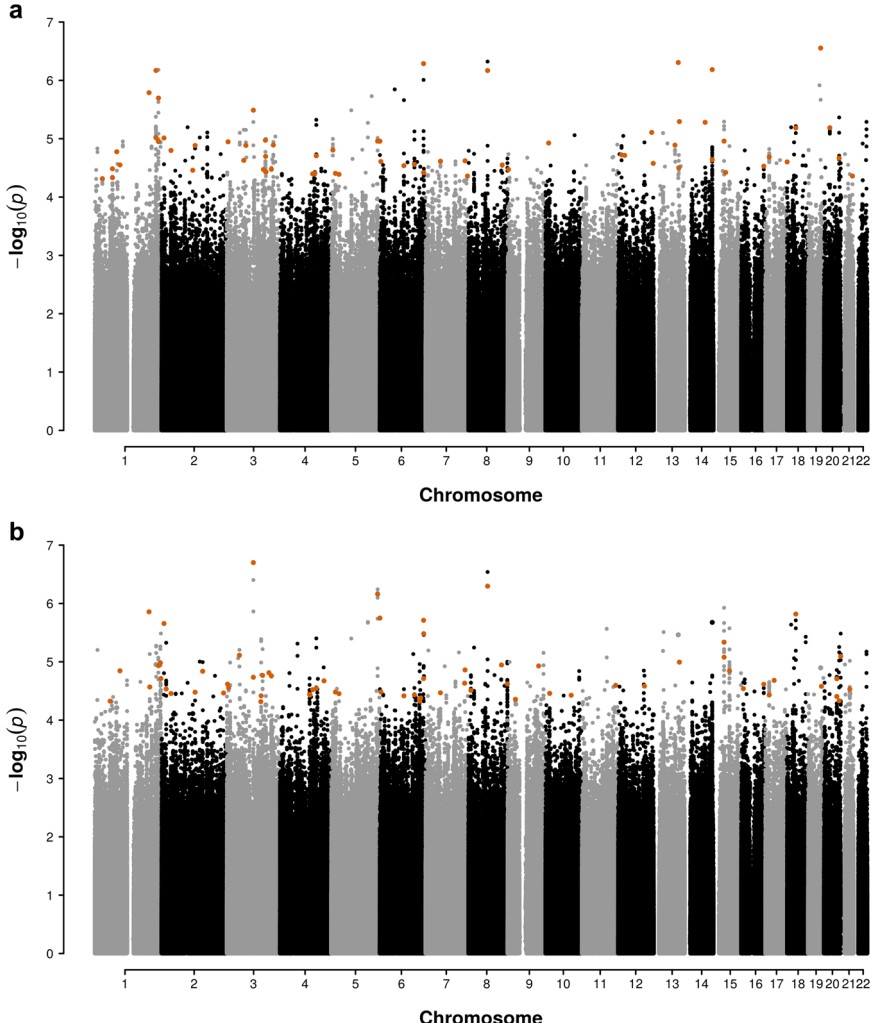

**Fig. 2 Manhattan plots of a GWAS of susceptibility to Buruli ulcer in a population from Benin.** Distribution of observed $P$ values for association between genetic variants and Buruli ulcer considering either a binary (affected/unaffected; **a**) or a censored (age at onset for Buruli ulcer patients and age at examination for the exposed controls; **b**) phenotype. The effects of variant genotypes on Buruli ulcer were tested under an additive genetic model, using logistic regression and Cox models for the binary and censored phenotypes, respectively. The $-\log_{10}(P$ values) ($y$-axis) for association are presented according to chromosomal positions ($x$-axis). Larger orange dots correspond to the 105 variants selected for replication.

**Table 2 Association results for the two replicated Buruli ulcer GWAS signals.**

| | | | | | Discovery | Replication | Combined sample | | | |
| | | | | | | | | | | |
| SNP | Chr | Position[a] | Gene | M/m[b] | $P$ value[c] | $P$ value[c,d] | MAF Controls | MAF Cases | $P$ value[c] | Effect size[e] |
|---|---|---|---|---|---|---|---|---|---|---|
| rs9814705* | 3 | 145680345 | ENSG00000240095.1 | T/C | $1.67 \times 10^{-4}$ | $6.51 \times 10^{-4}$ | 0.10 | 0.16 | $2.85 \times 10^{-7}$ | 1.80 (1.43–2.27) |
| rs76647377 | 6 | 985196 | LINC01622 | G/A | $1.78 \times 10^{-6}$ | $3.26 \times 10^{-3}$ | 0.05 | 0.02 | $9.85 \times 10^{-8}$ | 0.41 (0.28–0.60) |

[a]GRCh37.p13.
[b]Allele $m$ is the minor allele and $M$ is the major allele in the combined cohort.
[c]Discovery, replication and combined $P$ values obtained when the logistic model (Buruli ulcer per se phenotype) is considered for rs9814705, and when the Cox model (taking age at onset into account) is considered for rs76647377. When considering mixed-models in the discovery sample, i.e. GEMMA for rs9814705 and coxme for rs76647377, discovery $P$ values are $2.13 \times 10^{-4}$ and $2.06 \times 10^{-6}$, respectively.
[d]One-tailed test.
[e]Effect size corresponds to odds ratios (95% confidence interval) when the logistic model (Buruli ulcer per se phenotype) is considered (rs9814705), and hazard ratios (95% confidence interval) when the Cox model (taking age at onset into account) is considered (rs76647377). Effect size is assessed under an additive model, using the minor allele as the reference.
*rs9814705 was resequenced in the discovery sample.

the replication $P$ value was $3.26 \times 10^{-3}$. When merging our two samples, the combined $P$ value for the association between rs76647377 and Buruli ulcer was $9.85 \times 10^{-8}$. The minor allele A (MAF = 0.03) was protective and the HR for developing Buruli ulcer for AA *vs.* GA or GA vs. GG carriers was estimated at 0.41 [95%CI: 0.28–0.60] (Table 2). Stated differently, GG carriers are prone to develop Buruli ulcer at an earlier age than GA or AA

carriers (Fig. 5). Given that there was only one AA individual, who was a control, it is impossible to distinguish between an additive and a dominant effect for allele A. Following the same strategy as the one described above, screening of the region identified three SNPs displaying an $r^2 > 0.5$ with rs76647377: rs116809810 and rs11969790 ($r^2 = 0.58$, GWAS $P$ value = $4.55 \times 10^{-5}$ for both), and rs144839883 ($r^2 = 0.54$, GWAS $P$ value =

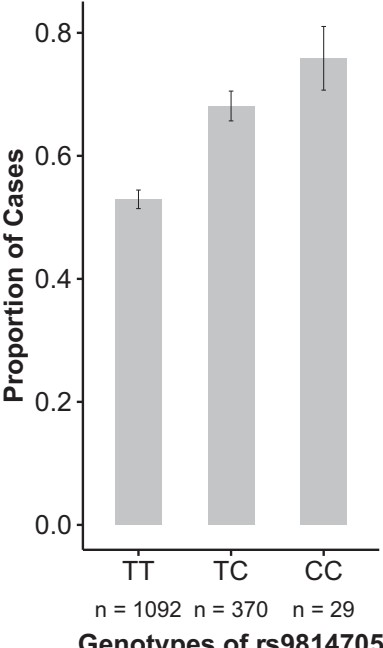

**Fig. 3 Genotype at rs9814705 is associated with the onset of Buruli ulcer.** Distribution of the proportion of Buruli ulcer cases (*y*-axis) according to rs9814705 genotype (*x*-axis) in the combined sample (*n*: number of individuals). Vertical bars represent standard errors.

$2.70 \times 10^{-3}$) (Fig. 6). To confirm the hypothesis of a single signal driving the association with the age of onset of Buruli ulcer in this chromosomal region, we performed association tests with all local variants conditioning on rs76647377. No variant in LD ($r^2 > 0.2$, shown in Fig. 6), including the aforementioned three SNPs, displayed *P* value < 0.05 supporting a single association signal accounted for by rs76647377 in this region (Supplementary Fig. 3b). Although we cannot decide for the causal variant on the basis of *P* values, it is important to note that this cluster of four SNPs in high LD ($r^2 > 0.5$) spans 9.7 kb in the intron of a LncRNA *LINC01622* of presently unknown function located on chromosome 6 (Table 2), with the closest reported protein-coding gene (*EXOC2*) located 292 kb away. Our results pinpoint this LncRNA as a good candidate for further investigations.

**Signals overlapping with previously reported associations.** As the microdeletion recently identified in a familial form of severe Buruli ulcer was close to a cluster of beta-defensin genes[20], we investigated the role of 5455 variants located in 50 defensin genes and/or identified as eQTL for defensin genes (Supplementary Data 2). No enrichment in *P* values either <0.01 or <0.001 was observed among these variants in our discovery sample, the best hit being rs58925751, located in the *DEFB123* gene on chromosome 20 (*P* value = $4.13 \times 10^{-4}$). We then checked the evidence of association in our GWAS for the six variants displaying significant association with Buruli ulcer in the three candidate-gene association studies on Buruli ulcer performed to date[25–27] (Table 3). Only one of these variants—rs2241880, a missense T300A variant located in the *ATG16L1* gene—had a *P* value < 0.05 in our GWAS under the additive model with the same allelic effect (one-tailed *P* value = 0.015). The association was even more significant (one-tailed *P* value = 0.009) when considering a recessive model for allele G (leading to the alanine aminoacid change), as reported in the original work. Under this recessive model, the OR for developing Buruli ulcer for GG vs. GA or AA carriers was estimated at 0.46 [95%CI: 0.25–0.86]. Furthermore,

in the previous study[26], GG homozygotes were found to be protected against the ulcerative clinical form of Buruli ulcer, with an OR of 0.35 [95%CI: 0.13–0.90]. Strikingly, when focusing on ulcerative cases of our discovery sample, the association was also more significant (*P* value = 0.003; OR = 0.31 [95%CI: 0.14–0.68]) despite the smaller sample size (255 ulcerative cases *vs.* 401 controls) (Table 3).

Finally, we also investigated whether some SNPs found to be associated with the other two most common mycobacterial diseases, tuberculosis and leprosy, overlapped with the SNPs identified in our GWAS on Buruli ulcer (see the "Methods" section for details). We first looked at the 105 SNPs selected for replication in our GWAS, and noted that rs9295218—the strongest hit among the genotyped SNPs identified in our discovery sample (GWAS *P* value = $5.17 \times 10^{-7}$; OR = 1.64 [95% CI: 1.35–2.00])—was located in an intron of the *PACRG* gene, variants of which have been associated with leprosy[29] (Supplementary Data 1). We observed the same trend towards association in the replication sample (replication *P* value = 0.13, with an OR of 1.14 [95%CI: 0.91–1.42]). This variant is not in LD with variants of the *PRKN* (a.k.a. *PARK2*)/*PACRG* cluster associated with leprosy, including that reported to be associated with Buruli ulcer by Capela et al.[26] ($r^2 = 0.007$) (Table 3). We then looked at the 35 SNPs reaching genome-wide significance in the GWAS performed on tuberculosis (eight independent variants in eight different chromosomal regions; Supplementary Data 3a) or leprosy (27 variants in 19 different chromosomal regions; Supplementary Data 3b). Using a type-I error of 0.01, none of these 35 SNPs was found to be significantly associated with Buruli ulcer. The lowest *P* value (0.017) was obtained for rs2269497, a missense variant of *RGS12* identified in a GWAS on tuberculosis in a Chinese population[30], for which the minor G allele was found to be associated with a risk of tuberculosis, whereas we found it to have a protective effect in Buruli ulcer. Taken together, these results suggest that there is no straightforward overlap between common variants potentially involved in the development of Buruli ulcer and in that of tuberculosis or leprosy.

## Discussion

Less than 3% of GWAS to date have focused on sub-Saharan African populations[31]. The inaccessibility of many areas makes it difficult to set up GWAS focusing on a neglected tropical disease. In this context, to the best of our knowledge, we report the first GWAS investigating susceptibility to Buruli ulcer in a well-characterized sample, including more than 1500 individuals, in Benin. This sample may be smaller than those used in GWAS on more common infectious diseases, such as leprosy and tuberculosis, but it is the largest sample ever used for a genetic association study on Buruli ulcer. Assuming a type I error of $5 \times 10^{-5}$ (the threshold for a variant to be considered for replication), the discovery sample had a power of 80% for detecting variants with a MAF > 0.18 and an OR > 1.80 under the additive genetic model. Individuals were recruited for this study from the villages of Ouémé and Plateau, two areas of Benin in which Buruli ulcer is highly endemic (Fig. 1). The high local prevalence in these areas, and our study design, in which the controls were older than the cases, maximize the chances of controls having been exposed to *M. ulcerans*. The enrollment of unexposed controls would decrease the power of the study. In addition, the definition of Buruli ulcer for the discovery sample was based strictly on the laboratory confirmation of cases, increasing the quality of the criteria used for selecting cases and controls in this study. The geographic clustering of *M. ulcerans* and its slow substitution rate suggest that patients from a given area are likely to be

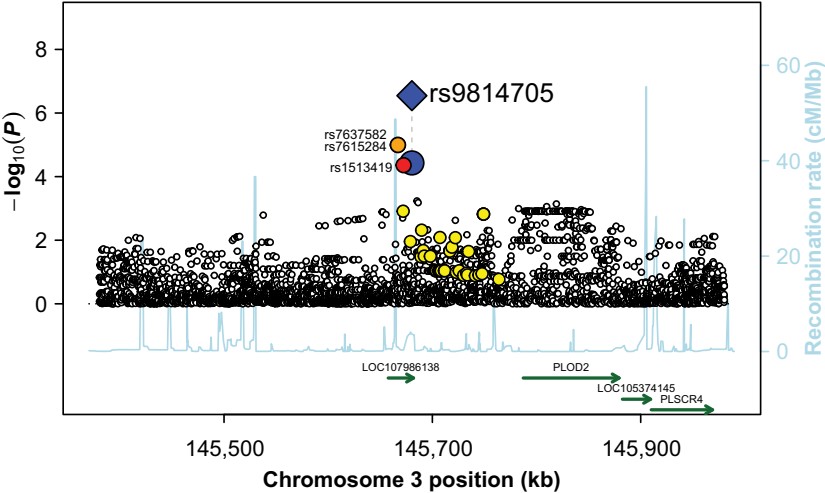

**Fig. 4 Regional linkage disequilibrium plot for rs9814705.** Evidence of association with Buruli ulcer ($-\log_{10}(P)$; left y-axis) for the variants located in the vicinity of rs9814705 (kb; x-axis). The distribution of recombination rates in this region is also given (cM/Mb; Right y-axis; light blue line). For the replicated SNP (i.e., rs9814705), the blue diamond corresponds to the result observed in the combined analysis, whereas the blue circle corresponds to the result observed in the GWAS only. A color-coded scheme is used to display the degree of LD of the proximal variants with the replicated SNP (red: $r^2 \geq 0.8$, orange: $0.5 \leq r^2 < 0.8$, and yellow: $0.2 \leq r^2 < 0.5$). Only variants with an $r^2 > 0.5$ are labeled. Known genes in the chromosomal region are plotted, with arrows indicating their orientation.

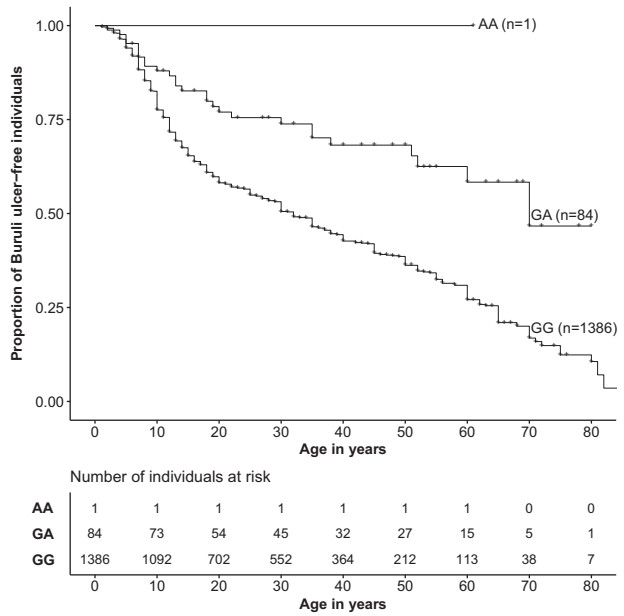

**Fig. 5 Genotype at rs76647377 is associated with age at onset of Buruli ulcer.** Kaplan–Meier curves for the onset of Buruli ulcer (y-axis) according to rs76647377 genotype. Number of individuals at risk are provided by genotype and by ten-year age bins.

infected by the same, or at least very similar *M. ulcerans* strains[32,33]. Strain variability, if any, would decrease the power of our GWAS, but not generate spurious signals of association.

Our study identified two independent loci associated with the onset (rs9814705, $P$ value $= 2.85 \times 10^{-7}$) or the age at onset (rs76647377, $P$ value $= 9.85 \times 10^{-8}$) of Buruli ulcer. These two variants are located in regions of the genome devoid of protein-coding genes: *PLOD2* is 107 kb away from rs9814705 and *EXOC2* is 292 kb away from rs76647377. Moreover, as expected in a sub-Saharan population, the LD pattern was of no help to resolve this issue. Both variants are located within the introns of LncRNAs: *ENSG00000240095.1* (rs9814705) and *LINC01622* (rs76647377).

Unfortunately, so far very limited functional data are available for *ENSG00000240095.1* and to the best of our knowledge none for *LINC01622*. Two correlated SNPs, rs9814705 and rs1513419 ($r^2 = 0.87$ with rs9814705), are eQTL for LncRNA *ENSG00000240095.1* in esophagus-mucosa tissue ($P$ value $= 3.9 \times 10^{-5}$ and $P$ value $= 4.5 \times 10^{-5}$, respectively)[34]. In the GTEx data, this LncRNA appeared to be expressed only in esophagus-mucosa tissue and in the minor salivary gland. In addition, rs9814705 and rs1513419 had $P$ values of borderline significance (0.049) for being cis-eQTL of *PLSCR1* in lipopolysaccharide-stimulated monocytes from healthy individuals[35]. The *PLSCR1* gene, located 565 kb away from rs9814705, encodes the transmembrane protein phospholipid scramblase 1, which has been shown to be overexpressed after stimulation with type I interferons, and to play a crucial role in apoptosis[36,37]. This gene has also been identified as a core gene in the host response to tuberculosis in a meta-analysis of transcriptomic data[38]. Thus, *PLSCR1* may be an interesting candidate gene for involvement in the development of Buruli ulcer. Further studies are required to determine the precise role of the identified LncRNAs.

Using our GWAS data, we replicated the effect of the A/G SNP rs2241880, located in the *ATG16L1* gene, initially identified in another sample from Benin[26]. Individuals homozygous for the minor allele G (MAF of 0.33 in the African populations of gnomAD) are protected against Buruli ulcer, especially the ulcerative forms, with an OR of 0.31 [0.14–0.68] in our sample and 0.35 [0.13–0.90] in the original study. The protein encoded by *ATG16L1* is required for autophagy[39], and the G allele has been associated with an increase in the release of cytokines, such as IL-1β and IL-18, leading to a decrease in antibacterial autophagy[40,41]. This variant results in a T300A missense substitution in the principal isoform of *ATG16L1*. Remarkably, it has also been reported to be an eQTL in nine tissues, including skin, in which the G allele of *ATG16L1* is associated with higher levels of expression of this gene ($P$ value $= 1.6 \times 10^{-25}$)[34]. Allele G of rs2241880 was also found to increase the risk of Crohn's disease (CD) in several GWAS, with an OR of about 1.4 under an additive model[42–44]. In addition, *Atg16l1*-hypomorphic mice developed intestinal abnormalities resembling CD whilst displaying resistance to intestinal infection with the bacterium

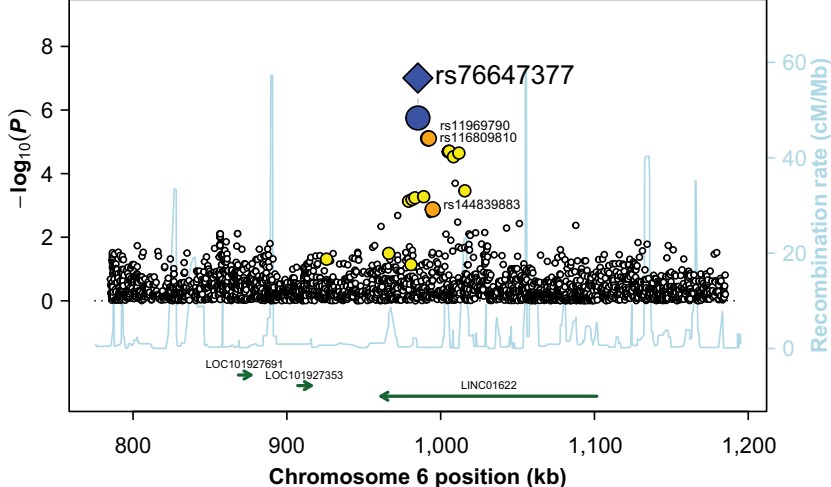

**Fig. 6 Regional linkage disequilibrium plot for rs76647377.** Evidence of association with Buruli ulcer ($-\log_{10}(P)$; left y-axis) for the variants located in the vicinity of rs76647377 (kb; x-axis). The distribution of recombination rates in this region is also given (cM/Mb; Right y-axis; light blue line). For the replicated SNP (i.e. rs76647377), the blue diamond corresponds to the result observed in the combined analysis, whereas the blue circle corresponds to the result observed in the GWAS only. A color-coded scheme is used to display the degree of LD of the proximal variants with the replicated SNP (red: $r^2 \geq$ 0.8, orange: $0.5 \leq r^2 < 0.8$, and yellow: $0.2 \leq r^2 < 0.5$). Only variants with an $r^2 > 0.5$ are labeled. Known genes in the chromosomal region are plotted, with arrows indicating their orientation.

**Table 3 Validation study for six variants located within six genes previously reported to be associated with Buruli ulcer.**

| SNP | Chr | M/m | Gene | Previous candidate-gene studies | | | | | | GWAS[a] | | |
|---|---|---|---|---|---|---|---|---|---|---|---|---|
| | | | | Study | Population | MAF | Model | OR (95% CI) | P value | MAF | OR (95% CI) | P value[b] |
| rs1333955 | 6 | **C**/T | PARK2 | Capela et al.[26] | Benin | 0.24 | Dom | 1.43 (1.00–2.06) | 0.05 | 0.27 | 1.05 (0.64–1.69) | 0.357 |
| rs9302752 | 16 | **C**/T | NOD2 | Capela et al.[26] | Benin | 0.38 | Dom | 2.23 (1.14–4.37) | 0.02 | 0.40 | 1.10 (0.72–1.67) | 0.498 |
| rs2241880 | 2 | A/**G** | ATG16L1 | Capela et al.[26] | Benin | 0.30 | Rec | 0.35 (0.13–0.90) | 0.02 | 0.26 | 0.46 (0.25–0.86) | **0.009** |
| rs9282799 | 17 | G/**A** | iNOS | Bibert et al.[27] | Ghana | 0.08 | Add | 1.99 (1.22–3.26) | 0.006 | 0.04 | 1.33 (0.82–2.14) | 0.110 |
| rs2069705 | 12 | A/**G** | IFNG | Bibert et al.[27] | Ghana | 0.47 | Add | 1.56 (1.14–1.99) | 0.007 | 0.43 | 1.00 (0.82–1.22) | 0.482 |
| rs17235409 | 2 | G/**A** | SCL11A1 | Stienstra et al.[25] | Ghana | 0.07 | Dom | 2.50 (1.26–4.96) | 0.007 | 0.09 | 1.02 (0.20–5.08) | 0.257 |

*Chr* chromosome, *M* major allele, *m* minor allele, *MAF* minor allele frequency, *Dom* dominant model, *Rec* recessive model, *Add* additive model.
[a]Results from the GWAS sample obtained with the binary case/control phenotype under the same genetic model as the one used by the corresponding previous study.
[b]P values for one-tailed tests are given for the marker alleles indicated in bold.

*Citrobacter rodentium*[45,46]. Published findings suggest that autophagy may serve as a rheostat for immune reactions[47]. These opposite effects in Buruli ulcer and CD provide another example of mirror genetic effects (i.e. an allele protective against infection but deleterious for inflammatory disorders), consistent with the view that the current high frequency of inflammatory/auto-immune diseases may reflect past selection for strong immune responses to combat infection[48].

Despite important practical and methodological challenges[49–51], we successfully conducted a GWAS on a neglected tropical disease in rural sub-Saharan Africa (Fig. 1), discovered two loci associated with Buruli ulcer and replicated a previously identified locus. Further studies are required to decipher the role of these variants and the function of the LncRNAs in which they are located. Detailed transcriptomic and epigenetic profiles for both whole blood (systemic response) and at the site of ulcers (local response) would complement our genomic approach and highlight genes and pathways involved in the disease. Mouse models of Buruli ulcer[52] will also be of major interest in this context, for further investigation of the role of the *ATG16L1* variant in the response to *M. ulcerans*, in particular.

## Methods
**Ethics statement.** This study on human genetic susceptibility to Buruli ulcer was approved by the Ethics Committee of the CDTLUB (Pobè, Benin), the International Review Board of the Ministry of Health in Benin (IRB00006860), and the

Ethics Committee of the University Hospital of Angers, France (Comité d'Ethique du CHU d'Angers). Written informed consent was obtained from all adult participants. Parents or guardians provided informed consent on behalf of all minors participating in the study.

**Subjects.** Between July 2003 and June 2012, 1524 individuals from the CDTLUB in Pobè, Benin, were enrolled. All participants were living in villages distributed over the Ouémé and Plateau *départements* (an administrative area equivalent to a county) in an area in which Buruli ulcer is endemic[53,54]. Based on the cases registered in these two areas between 2005 and 2012, the prevalence in the studied villages was 5.3 per 1000; a detailed distribution by village of the cases recorded is shown in Fig. 1. The discovery sample consisted of 408 HIV-free Buruli ulcer cases and 408 exposed controls (354 male and 462 female subjects; the mean ages of the cases and controls were 18.5 and 39.8 years, respectively). The replication sample consisted of 708 individuals: 467 HIV-free Buruli ulcer patients and 241 exposed controls (303 male and 405 female subjects; the mean ages of cases and controls were 22.3 and 26.1 years, respectively). To limit the risk of misclassification of controls, we sampled controls living in the same endemic areas as cases, and who were older than the cases. As the population under study is sedentary, the older the individuals, the longer they have been exposed to *M. ulcerans* still remaining Buruli ulcer-free. The vast majority of cases were confirmed through the detection of *M. ulcerans* DNA by polymerase chain reaction (PCR) targeting the insertion element IS2404 which is the most sensitive and specific laboratory test currently available for the diagnostic of Buruli ulcer[55,56]. Two cases had positive culture examination only. Overall, 98.3% of cases in the discovery sample and 45% in the replication sample were laboratory-confirmed through PCR and/or culture examination. All the cases that were not laboratory-confirmed were classified as highly probable on the basis of stringent clinical criteria as defined by the WHO[4]. Therefore, the risk of misclassification of cases lacking laboratory confirmation was likely very low. This is best exemplified by a recent study estimating that in Buruli ulcer-endemic setting, the ability of trained clinicians to diagnose Buruli ulcer on the basis of clinical

criteria had a sensitivity of 92% [95%CI: 85–96%] and a specificity of 91% [95%CI: 81–96%][57]. These estimates are particularly relevant in the context of our study as the CDTLUB was the main center of the abovementioned study providing approximately half of the total number of patients[57].

**Genotyping, imputation and quality control**. Genomic DNA was extracted from 5 to 10 mL blood samples with the Nucleon BACC2 Genomic DNA extraction kit (GE Healthcare), in accordance with the manufacturer's instructions. DNA preparations from the discovery sample were randomized in nine 96-well plates (with an equal number of cases and controls per plate) and genotyped with the Illumina Omni2.5 chip array. Genotypes were assessed with GenTrain implemented in Illumina GenomeStudio software (v2011.1). Thirteen of the 816 samples were excluded from further analyses due to a call rate <95% (eight samples), lack of concordance between reported and genetically inferred gender (three samples), or duplication (two samples). Our 'effective' discovery sample, therefore, consisted of a total of 803 individuals: 402 cases and 401 controls (Table 1).

In total, out of the 2,314,174 genotyped autosomal variants, a subset of 1,804,366, with a call rate >95%, a MAF > 0.01, and a Hardy–Weinberg $P$ value > $10^{-6}$ in the group of controls were used for the two-step (phasing and genotype inference) imputation process. Haplotypes were phased with SHAPEIT2 v2.904[58]. For non-genotyped variants, genotype was inferred with IMPUTE2 v2.3.2[59], using the 1000 Genomes Phase III dataset as the reference panel and default values for all other parameters[60,61]. Only variants imputed with an info value[59] above 0.6 were retained for further analysis. In total, 10,014,109 autosomal variants with a MAF above 0.02 if genotyped (1,683,899 variants) or 0.05 if imputed (8,330,210 variants) were tested for association with Buruli ulcer in our discovery sample.

A type I error of less than $5 \times 10^{-5}$ (genotyped variants) and $10^{-6}$ (imputed variants) was used to select variants for further genotyping and association testing in the replication sample. As the replication sample was derived from exactly the same population as the discovery sample, we were able to prune all the significant variants on the basis of their estimated LD to limit the genotyping effort. Among the variants of a given bin, defined as a group of variants with an $r^2 > 0.8$ with a core variant, we first selected the genotyped variant with the highest likelihood of successful genotyping in the replication sample, according to the manufacturer's instructions (see below). For bins without a genotyped variant or with genotyped variant for which the chances of genotyping success were low, we selected the imputed variant with the highest info value. Following LD pruning, 105 variants (99 genotyped and 6 imputed) were genotyped and tested for association with Buruli ulcer in the replication sample. These variants were genotyped by Illumina GoldenGate genotyping with VeraCode technology. Variants with a call rate < 95% or a Hardy–Weinberg $P$ value below $10^{-4}$ in controls were excluded. Among the 105 selected variants, five were excluded. Three of these variants were imputed in the discovery sample, two of which were present in Alu sequences, including rs7246288 which was the best hit in the GWAS sample, and the other in an LRT element. The two remaining variants were genotyped in the discovery sample but were found to be in Hardy–Weinberg disequilibrium among the controls of the replication sample ($P$ value < $10^{-4}$) (Supplementary Data 1). The replication sample contained 708 individuals; 693 of these individuals (455 cases and 238 controls; Table 1) passed the quality control criteria as 4 samples with a call rate < 95%, and 11 duplicated samples were excluded.

As one of the replicated SNPs (rs9814705) was imputed in the discovery sample, we decided to perform Sanger sequencing on this SNP in the discovery sample to strengthen our conclusions. Sequences for the 803 samples were obtained with the Big Dye Terminator kit and a 3500xL automated sequencer from Applied Biosystems. Sequence files and chromatograms were inspected with GENALYS software[62]. Given the high concordance between imputation and Sanger sequencing data (~99%), genotypes inferred by the imputation process were used for the 143 missing Sanger-sequenced genotypes.

**Statistics and reproducibility**. PCA was performed on genotyped variants with a MAF > 0.05 to check for population structure, for the discovery sample and all the samples available from 1000 Genomes Project Phase III[63], with the ad hoc functions implemented in PLINK v1.9[64]. To check for relatedness between individuals, we estimated the IBS between all individuals of the discovery sample using high quality genotyped variants with a MAF > 5% by means of PLINK v1.9[64]. We identified 39 pairs of first degree relatives including 16 phenotypically concordant pairs (8 pairs of controls, 8 pairs of cases) and 23 phenotypically discordant pairs. We also identified 36 pairs of second-degree relatives including 17 concordant pairs (12 pairs of controls, 5 pairs of cases) and 19 discordant pairs. We investigated the possible influence of relatedness in our GWAS by the use of specific methods detailed in the next paragraph. By contrast, as only 105 variants were genotyped in the replication sample, we could not perform similar relatedness or PCA analysis in this sample. However, recruitment of individuals of the replication sample was performed in the same areas concomitantly with the discovery sample, and we expect both population structure and relatedness to be comparable between both samples.

Two statistical designs were considered for tests of the association between genetic variants and the occurrence of Buruli ulcer, depending on the phenotypic definition use. We first considered Buruli ulcer as a binary phenotype (affected/unaffected). A classical case/control design was used and all analyses were

performed within a logistic regression framework. Gender had no significant effect ($P$ value = 0.41) on Buruli ulcer status. Given that the controls were chosen to be older than the cases by design, we did not include age as a covariate, to avoid over-adjustment. Instead, effect of age was accounted for by specific survival analysis methods (see below). We also evaluated the potential impact of relatedness and cryptic population stratification within our discovery sample, by comparing the results of the association tests obtained with a classical logistic regression model and those obtained with a mixed-effect logistic model[65]. We observed a very significant correlation ($r = 0.90$, Spearman $P$ value < $10^{-16}$) between the top 1000 variants obtained by classical logistic regression and those obtained with a mixed model (Supplementary Fig. 4a). The largest difference for these top 1000 variants, measured as the absolute difference between the $-\log_{10}(P$ value) obtained in classical logistic regression and the one obtained with the mixed model, was less than 0.81. Consistent with these findings, we found no significant association between the three first principal components of the PCA and the occurrence of Buruli ulcer.

We then considered age at onset for Buruli ulcer patients and age at examination for the exposed controls as the phenotype of interest. A survival analysis framework was used and all analyses were performed with Cox models[66]. In this analysis, we considered the Buruli ulcer diagnostic as the event of interest, i.e. the age at diagnosis being the failure time, and the age at which controls were recruited being the censored time. This age is indeed a good proxy of the duration of exposure since Buruli ulcer is highly endemic and the population is sedentary in the villages where the recruitment took place. The analysis strategy was identical to that used for the binary phenotype, i.e. we evaluated the potential impact of relatedness and cryptic population stratification within the discovery sample by comparing the results of the association tests performed with a classical Cox model with those obtained with a mixed-effect Cox model[67]. Again, the correlation between the $P$ values of the 1000 top variants for the classical and mixed-effect Cox models was highly significant ($r = 0.97$, Spearman $P$ value < $10^{-16}$ (Supplementary Fig. 4b)). The largest difference within these top 1000 variants, measured as the absolute difference between the $-\log_{10}(P$ value) obtained in classical Cox analysis and the one obtained with the mixed-effect Cox analysis was less than 0.28. Consistently, the use of the first three principal components of the PCA had no impact on the results.

Altogether, the analyses performed above confirmed the very limited impact of relatedness and/or population structure on the association results observed in our discovery sample using either the logistic (Buruli ulcer per se phenotype) or the Cox (taking age at onset into account) models. This is likely explained by the balanced distribution of phenotypically concordant and discordant relative pairs, and the ethnic homogeneity of our population. Therefore, all the results reported in the manuscript are those for association tests performed with classical logistic or Cox models. All analyses were conducted assuming an additive genetic effect of the variants, and all the $P$ values displayed are the ones obtained by likelihood-ratio tests. All association analyses were performed with R (https://www.r-project.org/), PLINK 1.9[64], GEMMA[65], coxme[67], SNPTEST v2.5.2[68], and ProbABEL[69] software.

For the replication study, we defined as significant a variant having the same allelic effect under the same genetic (additive) model in the discovery sample, with a one-tailed $P$ value < 0.01 in the replication sample. Similar replication thresholds (0.01 or 0.05) have been used in previous GWAS after having selected a similar number of variants (~100) for replication[21,70,71], and appears as a reasonable trade-off to limit the risk of false positive and false negative results. In our analysis the final interpretation of a replicated variant was based on the $P$ value combining the results of the discovery and the replication samples that was compared to the classical genome-wide $5 \times 10^{-8}$ threshold.

Finally, we calculated the power of our discovery sample (402 cases and 401 controls) for detecting a significant association with Buruli ulcer coded as a binary trait assuming an additive genetic model and a type I error of $5 \times 10^{-5}$. This threshold is a pragmatic trade-off to limit the number of false positives and false negatives that was used before[70]. This calculation was performed under the assumption of absence of misclassification of controls. Although we chose controls older than cases and living in the same endemic areas, we cannot not completely rule out some degree of misclassification of controls that would lead to a reduced power. Power estimates are shown according to the MAF and effect size, as estimated by the OR of the tested variants (Supplementary Fig. 5). For example, our sample has 50% power for the detection of variants with an OR of 1.6 and a MAF > 0.2, and 80% power for detecting variants with an OR of 1.8 and a MAF > 0.18. All these power estimates were calculated with QUANTO v1.2.4[72].

**Validation of variants previously found to be associated with common mycobacterial diseases**. We used our discovery sample to assess genetic findings already reported for Buruli ulcer and common mycobacterial diseases, i.e. leprosy and tuberculosis. We recently identified a microdeletion in a familial form of Buruli ulcer that overlaps with a cluster of genes encoding defensins[20]. Therefore, we identified all the variants located within the 50 defensin genes (including those located within 10 kb on either side of the gene) listed in the Ensembl GRCh37.p13 database and tested for association in our GWAS. We also considered all the variants reported to be significant (FDR < 0.05) eQTL for these defensin genes in whole-blood or skin from the lower legs exposed to the sun in the GTEx database[34]. Results were available for 5455 variants, of which 5450 were located in 50

defensin genes and 91 were identified as eQTL for defensin genes (86 of these eQTL were also located within defensin genes ±10 kb) (Supplementary Data 2). We then searched for variants reported to be associated with Buruli ulcer in the three published candidate-gene studies[25–27]. A total of six variants on six genes were assessed. Finally, we investigated the role of host genetic factors potentially common to Buruli ulcer and the other two main mycobacterial diseases, i.e. tuberculosis and leprosy. We screened the GWAS catalog[73] for variants associated with either tuberculosis or leprosy at the genome-wide level ($P$ value $< 5 \times 10^{-8}$). We identified 35 SNPs, eight in TB, and 27 in leprosy, which we then assessed for association with Buruli ulcer in our discovery sample.

**Reporting summary**. Further information on research design is available in the Nature Research Reporting Summary linked to this article.

## Data availability

The data that support the findings of this study are available within the paper and its Supplementary Information files. GWAS summary statistics for the two phenotypes studied, and source data for generating Fig. 5 are available at https://doi.org/10.6084/m9.figshare.11941344. All other relevant data are available from J.M. (jeremy.manry@inserm.fr) and L.A. (laurent.abel@inserm.fr) upon request.

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

## Acknowledgements

We thank Marie Kempf and Jane Cottin (Laboratoire de Bactériologie, CHU d'Angers, Angers, France), Jean-Paul Saint-André (Laboratoire d'Anatomie Pathologique, CHU d'Angers, Angers, France), Ambroise Adeye and Aimé Goundote (CDTLUB, Pobè, Benin), and Jean Gabin Houezo and Didier Agossadou (Programme de Lutte Contre la Lèpre et l'Ulcère de Buruli, Ministère de la Santé, Cotonou, Benin). We thank Jean-Laurent Casanova, Emmanuelle Jouanguy and all the members of the HGID laboratory for useful discussions. We also thank Guillaume Vogt, Antoine Guérin, Jérémie Babonneau, Julien Guergnon, Grégoire Hure, Natasha Vladikine, and Myriam Berramdane for their contribution to genotyping and sequencing. We acknowledge support from the Fondation Raoul Follereau. LM and AA were supported by the Agence Nationale de la Recherche (ANR; grants no. ANR-12-BSV3-0013-01 and ANR-17-BSV3-0013-01). J.M., L.A., and A.A. were supported by the Laboratoire d'Excellence "Integrative Biology of Emerging Infectious Diseases" (grant no. ANR-10-LABX-62-IBEID) and the ANR under the "Investments for the Future" program (grant no. ANR-10-IAHU-01). QBV was supported by the Fondation Bettencourt Schueller through the MD/PhD program of the Imagine Institute. A.A. was supported by the Fondation pour la Recherche Médicale (grant no. DMI20091117308).

## Author contributions

All authors were responsible for the study concept and design, and the analysis and interpretation of data. M.-F.A., C.J., L.M., E.M., and A.C. collected the data. M.-F.A., L.M., E.M., and A.C. did or confirmed diagnostics. Q.V., J.M., M.F.A., E.M., C.J., A.C., M.C., L.L., I.T., and A.A. acquired the data. Q.V., J.M., L.A., and A.A. did the statistical analysis. J.M., L.A., and A.A. drafted the report. A.A. obtained the funding.

## Competing interests

The authors declare no competing interests.
