## [Peer Review File · Communications Biology]

Reviewers' comments:

Reviewer #1 (Remarks to the Author):

Dr. Manry and colleagues present an interesting article on an understudied disease in an understudied region of the world. This genome wide association study of Buruli ulcers in Benin provides the first genome wide evaluation of genes that may influence infectious outcomes. Buruli ulcers appear to be growing in prevalence especially in West Africa warranting attention on potential risk factors including genetics.

The study is designed in 2 phases. The first is a case-control genetic study followed with a replication in separate group of cases and controls of the top variants. Plus the authors evaluate known mtb and leprosy genetic variants and other candidate genes.

There are some areas that need to be addressed to provide clarity and confidence.

1) A complete case definition is lacking. Especially, for this audience that is unlikely familiar with Buruli ulcers. How are cases determined? What is the clinical evaluation? How definitive is a diagnosis? And are the tests sensitive/specific to diagnosis? Similarly, how is exposure determined? Is this just being a certain age and living in that region? Or is there a test to say that an individual has been exposed to this mycobacteria but did not develop ulcers? how prevalent is exposure across these mapped regions?

Exposure is very important because otherwise there is the potential for misclassification in the controls. Please address.

2) Since both GWAS and replication were from the same population. Were there any related individuals among the cases and controls? The authors mention using mixed models, but I would prefer to see traditional relatedness (ϕ) or other measures to evaluate the relatedness of individuals. It can be addressed in the mixed models if found. Similarly, is there a relationship between cases and controls across the gwas and replication set? Can that be determined? It is plausible that all cases could be related even distantly?

3) The PCA plots are not very helpful. I would replot with just cases/controls so that we can see the spread between those cases and controls that could lead to population stratification. It's clear from PCA that they cluster with Africans and then with YRI in the African specific plots. But we can't even see the gray dots in 2a plot. Was any PCA done in replication? You need to address in paper if not--the reasons you do not think the replication is not inflated.

4) Was a conditional analysis performed to confirm that the top SNPs you identified represented the LD in the region? Since these are non-functional SNPs in genes of unknown function would be great to know that at least it represents other SNPs in each region. And, were there any other potential functional SNPs in these regions--even if they were not the top SNP? What happens if you condition on them?

Figure 1: the orange dots are not visible. Maybe make all the dots larger. Or just provide that information in a table.

Figure 2: Not sure this figure is necessary. It is unclear what it is adding overall.

Table 4: This table is appreciated but the table is a little confusing. Is the MAF from the previous study? and then the new MAF the MAF from your study? What about the OR and P value from your study? And did you replicate the same genetic model?

Reviewer #2 (Remarks to the Author):

The authors performed the first GWAS in Buruli ulcer, the third most frequent mycobacterial infectious disease. They identified two SNPs located near lncRNAs associated with susceptibility to Buruli ulcer. Another unexpected finding of the GWAS was the mirrored association of the ATG16L1 T300A amino acid change between Buruli and Crohn's disease. This finding further increased the evidence that past host adaptation to pathogens favored the selection of variants underlying susceptibility to immune-mediated diseases today. Last not least, the authors took the extra effort of evaluating their data using both linear and mixed models to address potential population stratification effects. Overall, these results are exciting since Buruli ulcer, from the genetic point of view, is a highly neglected disease even among the group of NTDs. Equally exciting is the link of the third mycobacterial disease, i.e. Buruli, with Crohn's disease. The implication of the latter link, especially between leprosy and CD, are an area of scientific controversy and the present paper provides important insight into this discussion.

Despite the above comments, I do have a few suggestions that the authors may want to consider.

Major

- The top SNPs in the discovery phase, rs7246288 for the binary phenotype, and rs34060873 for the Cox model were only mentioned once in the entire text. Do these SNPs fail to replicate their association with Buruli ulcer or other replicated SNPs tagged their association? This should be added.
- The rationale for the censored analysis needs to be better explained in the main text. Is there any evidence of age playing an important role in Buruli ulcer? If the authors adjust the binary analysis by age at disease onset, do they recover the association of rs76647377 with the binary phenotype?
- Among the SNPs selected for replication, how many were selected due to the binary phenotype or the cox model? What was the overlap of SNPs selected for replication between the binary and the cox model?
- A trend was observed for the association of rs6478108 of the TNFSF15 gene with Buruli ulcer (Table S3B). This SNP is associated with both CD and T1R in leprosy. In Buruli ulcer patients experience a host-immune mediated reaction similar to T1R in leprosy termed paradoxical reaction. Is it possible that rs6478108 trend of association with Buruli is tagging susceptible to paradoxical reactions? Do the authors have information regarding paradoxical reactions on their sample set to perform a stratified analysis?
- What is the genetic diversity of *M. ulcerans* strains? Is it high as for observed for *M. tuberculosis* or low as for *M. leprae*? Could *M. ulcerans* strain diversity impact on the GWAS results? This point needs to be addressed in the discussion. For instance, GWAS in leprosy were more successful than GWAS in TB in part due to the homogeneity of the bacteria infecting cases.
- A paragraph citing the limitations of the study is needed, in particular for the replication strategy. Since the authors used LD pruning before selecting markers for replication the majority of the SNPs selected were independent. Since 100 SNPs were selected, a cut off of 0.01 is not very stringent and this should be stated.

Minor

- Tables 2 and 3 and figure 2 could be summarized in a single table.
- The results shown in table 4 are confusing. Where are the p values and OR from the GWAS? In the table, only the original values were shown.
- The authors argue that common variants seem to play only a limited role in tuberculosis. While this may be correct, most of the TB studies did not account for strain diversity, limiting the interpretation of the studies in TB

Reviewer #3 (Remarks to the Author):

This manuscript is very well written, scientifically sound and provides novel insights into a largely neglected disease. I have no major comments, however I do have some minor suggestions as outlined below.

- Figure 4 is not discussed in the manuscript. Please address this.
- In the introduction, reference is made to rare and common variants however this is not touched on in the discussion. This is particularly noteworthy as the two association signals consisted of a rare and a common variant. Furthermore, there should be some revision of the last paragraph of the introduction, to round off the manuscript and tie in with what is mentioned in the discussion.
- Please include why a p value cut-off of 5×10^{-5} was used.
- Please include the imputation *info* value for the one association signal.
- Please confirm within the manuscript that the study participants were consented to not only take part in the research but that they consented to their de-identified data being made publicly available.

In this point-by-point response, we have reproduced the comments made by the reviewers. We have answered the questions in the order as they appear in the reviews and explain how we have changed the manuscript to take the suggestions made into account. We used the following color code throughout the response to reviewers to facilitate reading: reviewer questions in blue and italics, author answers in black. Within the answers, we highlighted the actions taken in bold, and the modified text with yellow background.

Reviewer #1 (Remarks to the Author):

Dr. Manry and colleagues present an interesting article on an understudied disease in an understudied region of the world. This genome wide association study of Buruli ulcers in Benin provides the first genome wide evaluation of genes that may influence infectious outcomes. Buruli ulcers appear to be growing in prevalence especially in West Africa warranting attention on potential risk factors including genetics. The study is designed in 2 phases. The first is a case-control genetic study followed with a replication in separate group of cases and controls of the top variants. Plus the authors evaluate known mtb and leprosy genetic variants and other candidate genes. There are some areas that need to be addressed to provide clarity and confidence.

We are grateful to the Reviewer for her/his helping comments and suggestions.

1) A complete case definition is lacking. Especially, for this audience that is unlikely familiar with Buruli ulcers.

We thank the Reviewer for bringing up the question of the diagnosis w.r.t. the readership. We agree that this question is of utmost importance, and we therefore substantially clarified this aspect of the manuscript (see below). For sake of clarity, point by point answers to diagnosis-related issues raised by the reviewer (1a to 1d) are given below.

1.a How are cases determined? What is the clinical evaluation? How definitive is a diagnosis? And are the tests sensitive/specific to diagnosis?

All cases were prospective cases. All cases were diagnosed in a unique, world renowned center dedicated to the diagnostic of Buruli ulcer (*Centre de Dépistage et de Traitement de la Lèpre et de l'Ulcère de Buruli*, CDTLUB). All cases were clinically diagnosed according to the latest WHO diagnostic criteria, i.e. young age, residence in an endemic area, undermined edges, location on limbs, necrosis, absence of pain, adenopathy or fever, and hyperpigmented edges (<https://apps.who.int/iris/handle/10665/274607>).

The vast majority of cases were laboratory-confirmed through the detection of *M. ulcerans* DNA by polymerase chain reaction (PCR) targeting the insertion element IS2404 on fine-needle aspiration, biopsy or swab samples. PCR is the most sensitive and specific laboratory test currently available for the diagnostic of Buruli ulcer¹. Some cases had also positive culture examination. Overall, 98.3% of cases in the discovery sample and 45.0% in the replication sample were laboratory-confirmed by PCR and/or culture examination.

For non-laboratory confirmed cases, only patients classified as 'highly probable' on the basis of the WHO diagnostic criteria described above including Ziehl-Neelsen staining were considered for the study. Accounting for the clinical expertise of the CDTLUB, we can safely assume the risk of misclassification as extremely limited. This is best exemplified by a recent study estimating that in Buruli ulcer-endemic setting, clinically-based diagnosis of Buruli ulcer by trained clinicians has a sensitivity of 92% and a specificity of 91%². These estimates are particularly applicable to the CDTLUB as it was the main center of the study providing ~ half of the total number of patients².

Overall, we used very stringent criteria for the diagnosis of Buruli ulcer and consider such stringency as a major strength of our study.

We added the following sentences at the end of the “Subject” section of the Methods regarding cases without laboratory confirmation:

“The vast majority of cases were confirmed through the detection of *M. ulcerans* DNA by polymerase chain reaction (PCR) targeting the insertion element IS2404 which is the most sensitive and specific laboratory test currently available for the diagnostic of Buruli ulcer^{1, 3}. Two cases had positive culture examination only. Overall, 98.3% of cases in the discovery sample and 45% in the replication sample were laboratory-confirmed through PCR and/or culture examination. All the cases that were not laboratory confirmed were classified as highly probable on the basis of stringent clinical criteria as defined by the WHO⁴. Therefore, the risk of misclassification of cases lacking laboratory confirmation was likely very low. This is best exemplified by a recent study estimating that in Buruli ulcer-endemic setting, the ability of trained clinicians to diagnose Buruli ulcer on the basis of clinical criteria had a sensitivity of 92% [95% CI:85%-96%] and a specificity of 91% [95% CI, 81%-96%]². These estimates are particularly relevant in the context of our study as the CDTLUB was the main center of the abovementioned study providing approximately half of the total number of patients².”

1.b Similarly, how is exposure determined? Is this just being a certain age and living in that region? Or is there a test to say that an individual has been exposed to this mycobacteria but did not develop ulcers?

Ideally, controls should be individuals proven to be infected by *M. ulcerans* but not developing the disease. There is currently no validated technique to identify such asymptomatic infected individuals. Similarly, there is no validated procedure to quantify exposure to the microbe.

Therefore, as fully understood by the reviewer, estimate of exposure was pragmatic. In practice, we enrolled controls living in the very same area as the cases, having similar habits and being older than cases to ensure highest prevalence of exposure to the microbe. We realize this is not perfect but to our knowledge this is the optimal procedure. As all of the controls were sedentary, ‘being older’ is a fair tag for remaining Buruli ulcer-free although exposed to *M. ulcerans* for a longer time than cases.

We thus clarified at the end of the “Subject” section of the Methods how controls were recruited:

“The replication sample consisted of 708 individuals: 467 HIV-free Buruli ulcer patients and 241 exposed controls (303 male and 405 female subjects; the mean ages of cases and controls were 22.3 and 26.1 years, respectively). To limit the risk of misclassification of controls, we sampled controls living in the same endemic areas as cases, and who were older than the cases. As the population under study is sedentary, the older the individuals, the longer they have been exposed to *M. ulcerans* still remaining Buruli ulcer-free.”

1.c how prevalent is exposure across these mapped regions?

For reasons explained in the above paragraph, there is no scientifically sound method to provide estimate of the prevalence of the exposure to *M. ulcerans*. Obviously it is higher than the prevalence of the disease but stating to which extent would be pure speculation. Because the cases were sampled in highly endemic areas as indicated in the map we provide in Supplementary Figure 1, we consider a fair assumption that exposure prevalence among controls to be at least as high as the one among cases.

1.d Exposure is very important because otherwise there is the potential for misclassification in the controls. Please address.

This statement is sound and correct. However, the impact of misclassification would be loss of power and not increase in false positive signals. **We have mentioned this issue in the revised version of the manuscript, in the “Statistical analyses” section of the Methods:**

“Finally, we calculated the power of our discovery sample (402 cases and 401 controls) for detecting a significant association with Buruli ulcer coded as a binary trait assuming an additive genetic model and a type I error of 5×10^{-5} .”... “This calculation was performed under the assumption of absence of misclassification of controls. Although we chose controls older than cases and living in the same endemic areas, we cannot not completely rule out some degree of misclassification of controls that would lead to a reduced power. Power estimates are shown according to the MAF and effect size, as estimated by the OR of the tested variants (Supplementary Fig. 6).”

2) Since both GWAS and replication were from the same population. Were there any related individuals among the cases and controls? The authors mention using mixed models, but I would prefer to see traditional relatedness (phat) or other measures to evaluate the relatedness of individuals. It can be addressed in the mixed models if found. Similarly, is there a relationship between cases and controls across the gwas and replication set? Can that be determined? It is plausible that all cases could be related even distantly?

This is another important point raised by the Reviewer and we share her/his concern. The sampling for both the GWAS and the replication samples was completed in small to medium-sized isolated villages, and some degree of relatedness between samples is likely. To answer this question, we used three different approaches.

First, we estimated relatedness by computing the identity by state (IBS) estimates between all individuals of the discovery sample using high quality genotyped variants displaying MAF > 5%. We identified 39 pairs of first degree relatives including 16 phenotypically concordant pairs (8 pairs of controls, 8 pairs of cases) and 23 phenotypically discordant pairs. We also identified 36 pairs of second degree relatives including 17 concordant pairs (12 pairs of controls, 5 pairs of cases) and 19 discordant pairs. Concordant pairs can increase the type I error, while discordant pairs can decrease power. The number of phenotypically concordant pairs being lower than the number of discordant ones, the overall impact of relatedness, if any, would be a loss of power and not increase of false positive association signals.

Second, we compared the observed distributions of P values in our GWAS when analyzing the data by means of either standard statistical tools or statistical tools designed to account for relatedness (generalized linear mixed model as implemented in GEMMA, and Cox mixed model as implemented in coxme^{5,6}). Overall, we observed very high correlation (Figure 1 below - Supplementary Figure 4 in our manuscript).

Figure 1: Correlation between P values obtained with general linear models vs. mixed models

Top 1,000 $-\log_{10}(P)$ values are plotted for (a) logistic regression: GLM (x-axis) vs. mixed model (GEMMA) (y-axis), $r=0.90$, and (b) survival analysis: Cox proportional hazards model (x-axis) vs. mixed-effect CoxME (y-axis), $r=0.97$ where r is Spearman's correlation coefficient. These findings validate the use of GLM and Cox models in the replication sample.

Notably, this high correlation is observed for the two top association signals reported in the manuscript:

rs9814705, GEMMA P value = 2.13×10^{-4} vs 1.67×10^{-4} with the standard logistic regression
rs76647377, CoxME P value = 2.06×10^{-6} vs 1.78×10^{-6} with the classical Cox model

Finally, we tested the association between the occurrence of Buruli ulcer and the three first principal components of the PCA and found no significant results.

Altogether these additional analyses show the limited impact, if any, of the relatedness of individuals on the results of our GWAS. Accordingly, the results reported in our manuscript do not include relatedness as a covariate. Because the number of variants genotyped in the replication is not sufficient to determine relatedness, it is important to note that only this strategy makes it possible to perform a valid combined analysis of the GWAS and the replication samples.

This important question raised by the reviewer has led to substantial additions to the ‘Statistical analyses’ section of the revised version of the manuscript

- 1) **we added the following sentences in the first paragraph of the “Statistical analyses” section of the Methods:**

“PCA was performed on genotyped variants with a MAF > 0.05 to check for population structure, for the discovery sample and all the samples available from 1000 Genomes Project Phase III ⁷, with the *ad hoc* functions implemented in PLINK v1.9 ⁸. To check for relatedness between individuals, we estimated the IBS between all individuals of the discovery sample using high quality genotyped variants with a MAF > 5% by means of PLINK v1.9 ⁸. We identified 39 pairs of first degree relatives including 16 phenotypically concordant pairs (8 pairs of controls, 8 pairs of cases) and 23 phenotypically discordant pairs. We also identified 36 pairs of second degree relatives including 17 concordant pairs (12 pairs of controls, 5 pairs of cases) and 19 discordant pairs. We investigated the possible influence of relatedness in our GWAS by the use of specific methods detailed in the next paragraph. By contrast, as only 105 variants were genotyped in the replication sample, we could not perform similar relatedness or PCA analysis in this sample. However, recruitment of individuals of the replication sample was performed in the same areas concomitantly with the discovery sample, and we expect both population structure and relatedness to be comparable between both samples.”

- 2) **We removed the words “hidden” and “or” in the second paragraph of the “Statistical analyses” section:**

“We also evaluated the possible impact of ~~hidden~~ relatedness and ~~or~~ cryptic population stratification within our discovery sample, by comparing the results of the association tests obtained with a classical logistic regression model and those obtained with a mixed-effect logistic model”.

- 3) **We added the following sentences at the beginning of the fourth paragraph of the “Statistical analyses” section of the Methods:**

“Altogether, the analyses performed above confirmed the very limited impact of relatedness and/or population structure on the association results observed in our discovery sample using either the logistic (Buruli ulcer *per se* phenotype) or the Cox (taking age at onset into account) models. This is likely explained by the balanced distribution of phenotypically concordant and discordant relative pairs, and the ethnic homogeneity of our population.”

- 4) **We added a footnote giving GEMMA and coxme P values in Table 2.**

“^c Discovery, replication and combined P values obtained when the logistic model (Buruli ulcer *per se* phenotype) is considered for rs9814705, and when the Cox model (taking age at onset into account) is considered for rs76647377. When considering mixed-models in the discovery sample, *i.e.* GEMMA for rs9814705 and coxme for rs76647377, discovery P values are 2.13×10^{-4} and 2.06×10^{-6} , respectively.”

3) The PCA plots are not very helpful. I would replot with just cases/controls so that we can see the spread between those cases and controls that could lead to population stratification. It's clear from PCA that they cluster with Africans and then with YRI in the African specific plots. But we can't even see the gray dots in 2a plot.

We agree with the reviewer and added the results of the PCA performed on the individuals used for our GWAS only.

These additional results have been included in several sections of the revised version of the manuscript:

- 1) panel “c” to Supplementary Figure 2 (Figure 2 here) considering only cases and controls, after exclusion of one individual of each 1st and 2nd degree relatives pair identified above.

Figure 2: Principal component analysis to study population stratification in the GWAS sample

- 2) We added the following sentence in the “Genome-wide analyses” section of the Results:

“Principal component analysis (PCA) on genotyped variants with a MAF > 0.05 revealed no evidence of population stratification in our sample, either at the global (Supplementary Fig. 2a) or African-specific (Supplementary Fig. 2b) level as all our individuals clustered with those of the Yoruba population of Ibadan, Nigeria (YRI) and the Esan population of Nigeria (ESN) from the 1000 Genomes Project. Refined PCA performed only on individuals from our study ruled out cryptic stratification as cases and controls were evenly distributed across the two first components (Supplementary Fig. 2c).”

We also completed the legend of Supplementary Fig. 2 with the following:

(c) First (x-axis) and second (y-axis) components of the refined PCA performed on the Buruli ulcer patients and the Buruli ulcer-free controls used in our GWAS after excluding one individual from each pair of first and second degree relatives.”

Was any PCA done in replication? You need to address in paper if not--the reasons you do not think the replication is not inflated.

For the same reason mentioned above for relatedness, i.e. the too low number of variants tested for replication, we could not perform PCA in the replication sample.

We thus added the following sentences at the end of the first paragraph of “Statistical analyses” section of the Methods to address both relatedness and population structure in the replication sample:

“By contrast, as only 105 variants were genotyped in the replication sample, we could not perform similar relatedness or PCA analysis in this sample. However, recruitment of individuals of the replication sample was performed in the same areas concomitantly with the discovery sample, and we expect both population structure and relatedness to be comparable between both samples.”

4) Was a conditional analysis performed to confirm that the top SNPs you identified represented the LD in the region? Since these are non -functional SNP in genes of unknown function would be great to know that at least it represents other SNPs in each region. And, were they are any other potential functional SNPs in these regions--even if they were not the top SNP? What happens if you condition on them?

We are grateful for this comment, and **thus performed a logistic regression conditioning on rs9814705 and conditioned the Cox models on rs76647377 to confirm both SNPs represent the LD in the corresponding regions.**

- 1) When conditioning on rs9814705, P values for the three SNPs displaying a strong LD with rs9814705 ($r^2 > 0.5$, association P values $< 10^{-4}$) become:

$$P_{rs7637582} = 0.05$$

$$P_{rs7615284} = 0.05$$

$$P_{rs1513419} = 0.39$$

These results confirm that the association signal observed in the region is well carried by rs9814705, and is similar to rs1513419 and explains a large proportion of the association signal carried by rs7637582 and rs7615284. Of note, both rs9814705 and rs1513419 are eQTL for the reported lincRNA *ENSG00000240095.1*, while nothing is reported for the two other SNPs.

When conditioning on rs9814705, no SNP in the region, nor on the entire chromosome 3, displays a notable decrease in association P value.

- 2) Likewise, when conditioning on rs76647377, all the variants with $r^2 > 0.2$ with this SNP and that displayed an association P value $< 10^{-4}$ prior conditioning now display P values > 0.05 , including the variants reported on Figure 3 of our manuscript:

$$P_{rs11969790} = 0.18$$

$$P_{rs116809810} = 0.18$$

$$P_{rs144839883} = 0.63$$

Regarding these analyses, we have made the following changes in the revised manuscript:

- 1) **We added the following sentences in the manuscript, in the Results section, “Replication study”,**
 - (i) **End of the second paragraph for rs9814705:**

“We then searched for additional variants in strong LD, i.e. $r^2 > 0.5$, with rs9814705, using the 1000 Genomes Phase III data for the Yoruba population. Three SNPs were identified: rs1513419 ($r^2 = 0.87$, GWAS P value = 4.37×10^{-5}), rs7637582 and rs7615284 ($r^2 = 0.69$, and GWAS P value = 1.01×10^{-5} for both) (Fig. 3a). To confirm the hypothesis of a single signal driving the association with the onset of Buruli ulcer in this chromosomal region, we performed association tests with all local variants conditioning on rs9814705 (i.e. rs9814705 was considered as a covariate in the analysis of the other variants). No variant in LD ($r^2 > 0.2$, shown in Fig. 3a) with rs9814705, including the three abovementioned SNPs, displayed P value < 0.05 consistent with a single association signal accounted for by rs9814705 in this region (Supplementary Fig. 4a). It is important to note that this cluster of four SNPs in high LD ($r^2 > 0.5$) spans 13.4 kb in a single intron of *ENSG00000240095.1*, a lincRNA of presently unknown function located on chromosome 3 with the closest identified protein-coding gene (*PLOD2*) located 107 kb away. While we cannot decide for the causal variant on the basis of P values, our results pinpoint this lincRNA as a solid candidate for further investigations.”

(ii) End of the third paragraph for rs76647377:

“Following the same strategy as the one described above, screening of the region identified three SNPs displaying an $r^2 > 0.5$ with rs76647377: rs116809810 and rs11969790 ($r^2 = 0.58$, GWAS P value = 4.55×10^{-5} for both), and rs144839883 ($r^2 = 0.54$, GWAS P value = 2.70×10^{-3}) (Fig. 3b). To confirm the hypothesis of a single signal driving the association with the age of onset of Buruli ulcer in this chromosomal region, we performed association tests with all local variants conditioning on rs76647377. No variant in LD ($r^2 > 0.2$, shown in Fig. 3b), including the aforementioned three SNPs, displayed P value < 0.05 supporting a single association signal accounted for by rs76647377 in this region (Supplementary Fig. 4b). Although we cannot decide for the causal variant on the basis of P values, it is important to note that this cluster of four SNPs in high LD ($r^2 > 0.5$) spans 9.7 kb in the intron of a lincRNA *LINC01622* of presently unknown function located on chromosome 6 (Table 2), with the closest reported protein-coding gene (*EXOC2*) located 292 kb away. Our results pinpoint this lincRNA as a good candidate for further investigations.”

- 2) For both SNPs, we plotted the association, LD and gene maps after conditioning on them (Figure 3 below), and added these two plots to a new Supplementary Figure 4a and 4b. (Thus, old Supplementary Figure 4 and 5 become Supplementary Figure 5 and 6, respectively).

Figure 3: Regional linkage disequilibrium plots after conditioning on the main associated SNPs

We added the following legend to the new Supplementary Figure 4:

“Supplementary Figure 4. Regional linkage disequilibrium plots after conditioning on the main associated SNPs. Residual evidence of association between Buruli ulcer and variants located in the vicinity of rs9814705 and rs9814705 according to coordinates in kb as provided in GRCh37 (x-axis).

Residual evidence of association is expressed as $-\log_{10}(P)$ of the association test after adjustment on rs9814705 (a) and rs9814705 (b). Recombination rates in cM/Mb are also given (Right y-axis; light blue line). A color-coded scheme is used to display the LD of the tested variants with the driving SNP (red: $r^2 > 0.8$, orange: $0.5 < r^2 < 0.8$, and yellow: $0.2 < r^2 < 0.5$). Known genes are also provided for each of the chromosomal regions with arrows indicating their orientation.”

Figure 1: the orange dots are not visible. Maybe make all the dots larger. Or just provide that information in a table.

According to the Reviewer’s suggestion, we made the orange dots larger of Figure 1 (Figure 4 below).

Figure 2: Not sure this figure is necessary. It is unclear what it is adding overall.

According to the comments of both Reviewers 1 and 2, and to reduce redundancy, we decided to keep Figure 2 as a main figure, and to pass Table 3 as a supplementary table. Table 3 is now Supplementary Table 2. Prior Supplementary Table 2 becomes Supplementary Table 3, and prior Supplementary Table 3a and 3b become Supplementary Table 4a and 4b.

Table 4: This table is appreciated but the table is a little confusing. Is the MAF from the previous study? and then the new MAF the MAF from your study? What about the OR and P value from your study? And did you replicate the same genetic model?

We deeply apologize to the three Reviewers. The Table 4 we provided was truncated and half of the information was missing (we sent Table 4 using portrait formatting instead of landscape formatting). **We corrected this. This table (new Table 3 in the revised version of our manuscript) now fulfils all the reviewers’ comments.**

Reviewer #2 (Remarks to the Author):

The authors performed the first GWAS in Buruli ulcer, the third most frequent mycobacterial infectious disease. They identified two SNPs located near lncRNAs associated with susceptibility to Buruli ulcer. Another unexpected finding of the GWAS was the mirrored association of the ATG16L1 T300A amino acid change between Buruli and Crohn's disease. This finding further increased the evidence that past host adaptation to pathogens favored the selection of variants underlying susceptibility to immune-mediated diseases today. Last not least, the authors took the extra effort of evaluating their data using both linear and mixed models to address potential population stratification effects. Overall, these results are exciting since Buruli ulcer, from the genetic point of view, is a highly neglected disease even among the group of NTDs. Equally exciting is the link of the third mycobacterial disease, i.e. Buruli, with Crohn's disease. The implication of the latter link, especially between leprosy and CD, are an area of scientific controversy and the present paper provides important insight into this discussion.

We thank the Reviewer for her/his positive impression on our work and her/his thorough and helping comments.

Despite the above comments, I do have a few suggestions that the authors may want to consider.

Major

• The top SNPs in the discovery phase, rs7246288 for the binary phenotype, and rs34060873 for the Cox model were only mentioned once in the entire text. Do these SNPs fail to replicate their association with Buruli ulcer or other replicated SNPs tagged their association? This should be added.

Concerning rs7246288, we failed to genotype this variant in the replication cohort. This variant was imputed in the GWAS sample. Genotyping failure could be explained by its location. Indeed, rs7246288 is located in an Alu sequence and such regions are highly prone to genotyping errors. In addition, this variant is now reported as tetra-allelic in the last GRCh38 reference. These are probably the reason why we could not find another SNP tagging rs7246288 outside the Alu sequence. It is thus very likely that the signal observed in the discovery phase was a false positive.

Concerning rs34060873, we genotyping was successful, however, the association signal in the replication sample was in the opposite direction as indicated in Supplementary Table 1.

According to the reviewer's suggestion, we added the following sentence in the "Genotyping, imputation and quality control" of the Methods of our manuscript:

For rs7246288, first paragraph: "These variants were genotyped by Illumina GoldenGate genotyping with VeraCode technology. Variants with a call rate < 95% or a Hardy Weinberg P value below 10^{-4} in controls were excluded. Among the 105 selected variants, five were excluded. Three of these variants were imputed in the discovery sample, two of which were present in Alu sequences, including rs7246288 which was the best hit in the GWAS sample, and the other in an LRT element. The two remaining variants were genotyped in the discovery sample but were found to be in Hardy-Weinberg disequilibrium among the controls of the replication sample (P value < 10^{-4}) (Supplementary Table 1). The replication sample contained 708 individuals; 693 of these individuals (455 cases and 238 controls; Table 1) passed the quality control criteria as 4 samples with a call rate < 95%, and 11 duplicated samples were excluded."

For rs34060873, we added a specific footnote in the Supplementary Table 1: “— corresponds to variants displaying association in the opposite direction with respect to the association in the discovery cohort, e.g. rs34060873, the best hit using the Cox model in the GWAS sample is likely a false positive.”

• *The rationale for the censored analysis needs to be better explained in the main text. Is there any evidence of age playing an important role in Buruli ulcer? If the authors adjust the binary analysis by age at disease onset, do they recover the association of rs76647377 with the binary phenotype?*

It has been shown Buruli ulcer is over-represented in children and that younger cases tend to be males (Vincent, Lancet Glob Health, 2014), and this observation made us investigate the age at onset using the Cox analysis. Of note, in the GWAS sample, we show that gender only has a minor impact on our results.

We thank the reviewer for stressing this out, and clarified the censored analysis in the main text, Methods’ section, “Statistical analyses” by the following sentences:

“We then considered age at onset for Buruli ulcer patients and age at examination for the exposed controls as the phenotype of interest. A survival analysis framework was used and all analyses were performed with Cox models⁹. In this analysis, we considered the Buruli ulcer diagnostic as the event of interest, i.e. the age at diagnosis being the failure time, and the age at which controls were recruited being the censored time. This age is indeed a good proxy of the duration of exposure since Buruli ulcer is highly endemic and the population is sedentary in the villages where the recruitment took place.”

Of note, rs76647377 was also captured with a P value $< 5 \times 10^{-5}$ in the binary analysis, but since the P value was lower using the Cox model, we decided to only consider this model for the replication study. The replication and combined P values remain lower under the Cox model compared to the logistic regression model. By design, we chose to recruit older controls compared to cases to increase the likelihood of controls for being strongly exposed to *M. ulcerans*. Consequently, adjusting by age would thus lead to over-adjustment. The only correct way to assess the age effect in our sample is through a survival method, such as the Cox model.

• *Among the SNPs selected for replication, how many were selected due to the binary phenotype or the cox model? What was the overlap of SNPs selected for replication between the binary and the cox model?*

We thank the Review for this comment. **The number of variants retained for replication for each analysis is now provided in the “Genome-wide analyses” section of the Results, and we now provide the information about the overlap in the main text, at the end of the “Genome-wide analysis” section of the Results:**

“Applying the same strategy as the one used for the binary phenotype, 68 variants were retained for genotyping and association testing in the replication sample (Supplementary Table 1). Among the 66 and 68 variants retained from the analysis of Buruli ulcer *per se* and the one taking age at onset into account, respectively, 29 displayed P values below our thresholds of selection in both analyses. For the replication analysis, the phenotypic definition providing the best P value for these variants was used. Altogether, analysis of the two phenotypic models led to the selection of 105 independent variants, 99 of which being genotyped (Supplementary Table 1).”

We now clearly identify these 29 variants in a footnote “g” in Supplementary Table 1.

• *A trend was observed for the association of rs6478108 of the TNFSF15 gene with Buruli ulcer (Table S3B). This SNP is associated with both CD and T1R in leprosy. In Buruli ulcer patients experience a host-immune mediated reaction similar to T1R in leprosy termed paradoxical reaction. Is it possible that rs6478108 trend of association with Buruli is tagging susceptible to paradoxical reactions? Do the authors have information regarding paradoxical reactions on their sample set to perform a stratified analysis?*

We thank the Reviewer for this highly interesting remark. We fully agree that rs6478108 trend of association with Buruli could tag susceptibility to paradoxical reactions. Unfortunately, we do not have the information about the paradoxical reaction status of the individuals of our sample.

• *What is the genetic diversity of *M. ulcerans* strains? Is it high as for observed for *M. tuberculosis* or low as for *M. leprae*? Could *M. ulcerans* strain diversity impact on the GWAS results? This point needs to be addressed in the discussion. For instance, GWAS in leprosy were more successful than GWAS in TB in part due to the homogeneity of the bacteria infecting cases.*

We thank the reviewer for this comment. **We added the following sentences to the first paragraph of the Discussion of our manuscript:**

“The high local prevalence in these areas, and our study design, in which the controls were older than the cases, maximize the chances of controls having been exposed to *M. ulcerans*. The enrollment of unexposed controls would decrease the power of the study. In addition, the definition of Buruli ulcer for the discovery sample was based strictly on the laboratory confirmation of cases, increasing the quality of the criteria used for selecting cases and controls in this study. The geographic clustering of *M. ulcerans* and its slow substitution rate suggest that patients from a given area are likely to be infected by the same, or at least very similar *M. ulcerans* strains^{10,11}. Strain variability, if any, would decrease the power of our GWAS, but not generate spurious signals of association.”

• *A paragraph citing the limitations of the study is needed, in particular for the replication strategy. Since the authors used LD pruning before selecting markers for replication the majority of the SNPs selected were independent. Since 100 SNPs were selected, a cut off of 0.01 is not very stringent and this should be stated.*

We agree that a threshold of 0.01 could seem to be not very stringent, but seems reasonable in a “discovery” study such a GWAS. Similar replication thresholds (0.01 or 0.05) have been widely used after having selected a similar number of variants (~100) for replication^{12, 13, 14}. In addition, choosing a less stringent threshold after a prior analysis was proposed by Roeder *et al.*¹⁵ by performing a linkage analysis prior to a GWAS. This is applicable with a GWAS followed by a replication. The rationale of choosing a higher *P* value threshold comes from the fact that we start with an *a priori* provided by the discovery sample. Association is observed in the GWAS sample, and this prior is used in the replication sample. In any case, the most important result to interpret the possible effect of a replicated SNP is the *P* value combining the results of the discovery and the replication samples, as this combined *P* value could be compared to the classical genome-wide significant threshold of 5×10^{-8} . In the present analysis, the combined *P* values of the two replicated SNPs were close to but remained above the genome-wide threshold.

More general limitations, which are even better defined in the revised version of our manuscript w.r.t. Reviewer #1’s suggestions, are now provided in the first paragraph of the Discussion.

To tackle the Reviewer’s suggestions,

- 1) **to clarify the choice of the 0.01 threshold in the replication, we added the following sentences to the last paragraph of the “Genotyping, imputation and quality control” section of the Methods:**

“Significant replication was defined as a variant having the same allelic effect under the same genetic (additive) model as in the discovery sample, with a one-tailed *P* value < 0.01 on the replication sample. Similar replication thresholds (0.01 or 0.05) have been used in previous GWAS after having selected a similar number of variants (~100) for replication^{12, 13, 14}, and appears as a reasonable trade-off to limit the risk of false positive and false negative results. In our analysis the final interpretation of a replicated variant

was based on the P value combining the results of the discovery and the replication samples that was compared to the classical genome-wide 5×10^{-8} threshold.”

2) **We extended the limitations of our study by adding of the following sentences at the end of the first paragraph of the Discussion:**

“The geographic clustering of *M. ulcerans* and its slow substitution rate suggest that patients from a given area are likely to be infected by the same, or at least very similar *M. ulcerans* strains^{10,11}. Strain variability, if any, would decrease the power of our GWAS, but not generate spurious signals of association.”

Minor

- *Tables 2 and 3 and figure 2 could be summarized in a single table.*

As we proposed to the first Reviewer, we decided to transfer Table 3 as Supplementary Table.

- *The results shown in table 4 are confusing. Where are the p values and OR from the GWAS? In the table, only the original values were shown.*

Again, we apologize: please, see answer to the first Reviewer about Table 4.

- *The authors argue that common variants seem to play only a limited role in tuberculosis. While this may be correct, most of the TB studies did not account for strain diversity, limiting the interpretation of the studies in TB*

We thank the Reviewer for this comment and totally agree with this statement.

Reviewer #3 (Remarks to the Author):

This manuscript is very well written, scientifically sound and provides novel insights into a largely neglected disease. I have no major comments, however I do have some minor suggestions as outlined below.

We thank the Reviewer for her/his positive impression on our work.

- Figure 4 is not discussed in the manuscript. Please address this.

We thank the Reviewer for seeing this issue and **added the following sentence to the manuscript, at the last paragraph of the “Replication study” section of the results:**

“When merging our two samples, the combined P value for the association between rs76647377 and Buruli ulcer was 9.85×10^{-8} . The minor allele A (MAF = 0.03) was protective and the HR for developing Buruli ulcer for AA vs. GA or GA vs. GG carriers was estimated at 0.41 [95%CI: 0.28-0.60] (Table 2). **Stated differently, GG carriers are prone to develop Buruli ulcer at an earlier age than GA or AA carriers (Fig. 4).**”

- In the introduction, reference is made to rare and common variants however this is not touched on in the discussion. This is particularly noteworthy as the two association signals consisted of a rare and a common variant. Furthermore, there should be some revision of the last paragraph of the introduction, to round off the manuscript and tie in with what is mentioned in the discussion.

We thank the Reviewer for this observation. GWAS genotyping array, by design, only focus on what are usually called “common variants”. Three types of variants according to their frequency: private variant (seen only once in the population), rare variant (minor allele frequency (MAF) < 1%) and common variant (MAF > 1%). The thresholds to define rare vs common variants are arbitrary and misleading. GWAS are focusing only on “common” variant due to the limited power for finding association with rarer variants. For example, regarding our study with our sample size, our power analysis shows very low power to highlight association with variant with MAF < 2%. That is why we decided to only keep variants with MAF > 2% for genotyped variants and MAF > 5% for imputed variants (since imputation is more prone to errors for variants with MAF < 5%¹⁶), which is the case for the two main hits we reported.

We added the following sentence at the end of the Introduction:

“Our analysis identified two novel variants independently associated with the onset of Buruli ulcer.”

- Please include why a p value cut-off of 5×10^{-5} was used.

There is not specific rule to define the threshold for selecting variants to send to replication. The threshold of 5×10^{-5} is a pragmatic trade-off to limit the number of false positives and false negatives that was used before¹², and depends on the sample size. Indeed, our power study shows that, at such threshold, we for example reach good power (80%) to find true associations with variants displaying MAF > 0.20 with OR > 1.8 (Figure 4).

Figure 4: Power estimates for the association study on the discovery sample

Power estimates are given for a type I error of 5×10^{-5} , a sample of 402 cases and 401 controls and an additive genetic model according to the MAF (x-axis) and effect size, as measured by the OR (color-coded curves).

We thus added the following sentence to last paragraph of the “Statistical analyses” section of the Methods:

“Finally, we calculated the power of our discovery sample (402 cases and 401 controls) for detecting a significant association with Buruli ulcer coded as a binary trait assuming an additive genetic model and a type I error of 5×10^{-5} . This threshold is a pragmatic trade-off to limit the number of false positives and false negatives that was used before¹². This calculation was performed under the assumption of absence of misclassification of controls. Although we chose controls older than cases and living in the same endemic areas, we cannot not completely rule out some degree of misclassification of controls that would lead to a reduced power. Power estimates are shown according to the MAF and effect size, as estimated by the OR of the tested variants (Supplementary Fig. 6). For example, our sample has 50% power for the detection of variants with an OR of 1.6 and a MAF > 0.2, and 80% power for detecting variants with an OR of 1.8 and a MAF > 0.18. All these power estimates were calculated with QUANTO v1.2.4¹⁷.”

- Please include the imputation info value for the one association signal.

We thank the Reviewer for pointing this out. We added the info value in the “Replication study” section of the Results:

“The first evidence of replication was observed for the binary phenotype and rs9814705 on chromosome 3, for which the replication P value was 6.51×10^{-4} . As this variant had been imputed in the discovery sample (imputation info value = 0.933), we decided to subject it to Sanger sequencing in the discovery sample.”

- Please confirm within the manuscript that the study participants were consented to not only take part in the research but that they consented to their de-identified data being made publicly available.

We thank the Reviewer for his/her concern about ethical issues. As indicated in the manuscript, patients consented to the research, however, consent did not include a statement about de-identified data being made publicly available. Therefore, genotypic data will not be deposited into a public database. However, they will be available upon request for collaborations.

References

1. Zingue D, Bouam A, Tian RBD, Drancourt M. Buruli Ulcer, a Prototype for Ecosystem-Related Infection, Caused by *Mycobacterium ulcerans*. *Clin Microbiol Rev* **31**, (2018).
2. Eddyani M, *et al.* Diagnostic Accuracy of Clinical and Microbiological Signs in Patients With Skin Lesions Resembling Buruli Ulcer in an Endemic Region. *Clin Infect Dis* **67**, 827-834 (2018).
3. Stienstra Y, *et al.* Analysis of an IS2404-based nested PCR for diagnosis of Buruli ulcer disease in regions of Ghana where the disease is endemic. *J Clin Microbiol* **41**, 794-797 (2003).
4. World Health Organization. Buruli ulcer (*Mycobacterium ulcerans* infection) Fact Sheet. (2018).
5. Therneau T. coxme: Mixed Effects Cox Models. R package version 2.2-3 [Internet]. 2012. Available from: <http://CRAN.R-project.org/package=coxme>. (2012).
6. Zhou X, Stephens M. Genome-wide efficient mixed-model analysis for association studies. *Nat Genet* **44**, 821-824 (2012).
7. Sudmant PH, *et al.* An integrated map of structural variation in 2,504 human genomes. *Nature* **526**, 75-81 (2015).
8. Chang CC, Chow CC, Tellier LC, Vattikuti S, Purcell SM, Lee JJ. Second-generation PLINK: rising to the challenge of larger and richer datasets. *Gigascience* **4**, 7 (2015).
9. Cox DR. Regression Models and Life-Tables. *Journal of the Royal Statistical Society Series B (Methodological)* **34**, 187-220 (1972).
10. Coudereau C, *et al.* Stable and Local Reservoirs of *Mycobacterium ulcerans* Inferred from the Nonrandom Distribution of Bacterial Genotypes, Benin. *Emerg Infect Dis* **26**, 491-503 (2020).
11. Vandelanootte K, *et al.* Multiple Introductions and Recent Spread of the Emerging Human Pathogen *Mycobacterium ulcerans* across Africa. *Genome Biol Evol* **9**, 414-426 (2017).
12. Patin E, *et al.* Genome-wide association study identifies variants associated with progression of liver fibrosis from HCV infection. *Gastroenterology* **143**, 1244-1252 e1212 (2012).
13. Thye T, *et al.* Genome-wide association analyses identifies a susceptibility locus for tuberculosis on chromosome 18q11.2. *Nat Genet* **42**, 739-741 (2010).
14. Zhang FR, *et al.* Genomewide association study of leprosy. *N Engl J Med* **361**, 2609-2618 (2009).
15. Roeder K, Bacanu SA, Wasserman L, Devlin B. Using linkage genome scans to improve power of association in genome scans. *Am J Hum Genet* **78**, 243-252 (2006).
16. Howie B, Fuchsberger C, Stephens M, Marchini J, Abecasis GR. Fast and accurate genotype imputation in genome-wide association studies through pre-phasing. *Nat Genet* **44**, 955-959 (2012).
17. Gauderman WJ. Sample size requirements for matched case-control studies of gene-environment interaction. *Stat Med* **21**, 35-50 (2002).

REVIEWERS' COMMENTS:

Reviewer #1 (Remarks to the Author):

The authors have done a great job at responding to my comments and the other reviewers. I am satisfied with these responses, and also think that the manuscript is much stronger. Overall, this is a neglected area of research on a neglected population (with regards to genetics especially). The authors have identified new genetic variants and now detail their case definitions and methods substantially that we can feel confident with their findings.

Reviewer #2 (Remarks to the Author):

The authors have addressed the points raised in my review extremely well. No further comments.

Reviewer #3 (Remarks to the Author):

Substantial revisions were performed and I thank the authors for taking the time to draft a well revised manuscript. I have no further major concerns.